# Insights on Chemical Crosslinking Strategies for Proteins

**DOI:** 10.3390/molecules27238124

**Published:** 2022-11-22

**Authors:** Brindha Jayachandran, Thansila N Parvin, M Mujahid Alam, Kaushik Chanda, Balamurali MM

**Affiliations:** 1Chemistry Division, School of Advanced Sciences, Vellore Institute of Technology, Chennai Campus, Vandalur-Kelambakkam Road, Chennai 600127, India; 2Department of Chemistry, College of Science, King Khalid University, P.O. Box 9004, Abha 61413, Saudi Arabia; 3Department of Chemistry, School of Advanced Sciences, Vellore Institute of Technology, Vellore 632014, India

**Keywords:** protein crosslinking, chemical crosslinkers, protein materials, biomaterials, drug delivery

## Abstract

Crosslinking of proteins has gained immense significance in the fabrication of biomaterials for various health care applications. Various novel chemical-based strategies are being continuously developed for intra-/inter-molecular crosslinking of proteins to create a network/matrix with desired mechanical/functional properties without imparting toxicity to the host system. Many materials that are used in biomedical and food packaging industries are prepared by chemical means of crosslinking the proteins, besides the physical or enzymatic means of crosslinking. Such chemical methods utilize the chemical compounds or crosslinkers available from natural sources or synthetically generated with the ability to form covalent/non-covalent bonds with proteins. Such linkages are possible with chemicals like carbodiimides/epoxides, while photo-induced novel chemical crosslinkers are also available. In this review, we have discussed different protein crosslinking strategies under chemical methods, along with the corresponding crosslinking reactions/conditions, material properties and significant applications.

## 1. Introduction

Protein crosslinking [1] that results in the formation of films/materials finds significant applications in food/non-food packaging [2,3] and health care industries [4,5]. Proteins, with their reactive side chains, interact with other protein or non-protein molecules, like polysaccharides or synthetic polymers, to generate crosslinked protein polymers [6] that can potentially replace the toxic non-degradable synthetic polymers [7]. Such crosslinking methods involve the formation of weak or strong bonds between the polypeptide chains within the protein (intramolecular crosslinks) or between different protein molecules (intermolecular crosslinks). This kind of crosslinking between proteins is significant as they possess the tremendous potential to be used in biomedical fields as scaffold materials, for drug delivery, or for other therapeutic purposes, along with the growing demands in food industries such as edible films, non-toxic packaging material and in coating applications [8,9]. With the availability of different kinds of proteins and characterization techniques [10], it is possible to develop protein-based biocompatible materials which can be used as biological scaffolds or matrices for various tissue engineering applications [11,12]. Protein-based materials provide an opportunity to generate a new era of biomaterials [13] with the ability to carry out versatile functions [14,15,16]. Proteins, with their unique features like biochemical/biomolecular interactions, self-assembling properties, and specific functional domains, along with their ability to be tuned at molecular scales [17], have attracted several researchers to explore protein crosslinking for various useful applications [18,19,20,21]. Several known strategies of protein crosslinking are majorly known to occur by targeting (i) the primary amines that include the N-terminal amines in each polypeptide chain and the one in the lysine side chain; (ii) the carboxyl groups in the C-terminus of each polypeptide chain and also the one in the aspartate and glutamate side chain; (iii) the sulfhydryl (-SH) group in the side chain of cysteine amino acid residues; and (iv) the carbonyl groups of carbohydrates in case of glycoproteins (Figure 1). In general, a crosslink denotes either a physical or chemical bond that associates the reactive groups in a polymer network [22] by means of stronger covalent bonding or weaker interactions like ionic/hydrogen bonding or other non-specific interactions. The crosslinking of proteins can be carried out by physical or chemical means or by using crosslinkers. The functional groups of the protein biomolecules are modified physically or chemically under specific conditions in the presence of crosslinkers, resulting in a protein network of different mechanical/functional properties. Physical means of crosslinking involve non-covalent/weak covalent bonds that are temporary or reversible, whereas chemical crosslinkers include natural/synthetic molecules that crosslink proteins with relatively stronger covalent bonds. Typically, the crosslinkers/crosslinking conditions are known to enhance the mechanical nature of the polymeric network formed, while it is also necessary that these crosslinkers do not impart toxicity. Crosslinking process also insolubilizes the protein polymeric network in water, forms materials/thin films and promotes the biomechanical stability of the crosslinked protein matrix under physiological conditions, which is important for its end use for biomedical applications [23].

In recent times, novel crosslinkers or crosslinking strategies have been developed as one of the promising research areas, focusing on the utilization of versatile protein biomolecules with potential biomedical and industrial applications [24]. The major methods of crosslinking include physical, irradiation-based, chemical and enzymatic [25], among which the enzymatic method is the most expensive [22] is not discussed here and could be referred to elsewhere [26]. Physical crosslinking of proteins is less expensive and relatively less cytotoxic when compared to other chemical methods, but it also has its own limitations of generating relatively weaker interactions upon crosslinking and possesses the possibility of altering the properties of proteins. Yet another class of crosslinkers involves irradiation-based methods that include UV (Ultra-violet) treatment, gamma irradiation and photo-chemical crosslinking techniques [27]. Whereas the chemical method of crosslinking is carried out using either naturally obtained crosslinkers or synthetically derived ones forming strong covalent bonds. These naturally derived chemical crosslinkers are usually non-cytotoxic, with a few forming weaker hydrogen bond cross-linkages between proteins, while the synthetically derived chemical crosslinkers form stronger covalent interactions with the limitation of possible cytotoxicity. From recent literature, it can be seen that attempts are being made to develop an ideal protein crosslinker either synthetically [28] or from natural sources [29] or using certain peptides to create stronger cross-linkages without causing cytotoxicity. This review focuses on different kinds of crosslinkers/crosslinking conditions for proteins that include significant chemical crosslinkers, along with the mechanism of crosslinking and potential applications.

## 2. Chemical Crosslinking of Proteins

Chemical crosslinking involves the utilization of different natural and/or synthetic crosslinkers to form the linkages. Protein crosslinkers usually possess reactive sites that target functional moieties like thiol, amine and carboxylic acid groups in the side chains of amino acid residues. The chemical crosslinkers can further be classified under two groups, namely, non-zero length crosslinkers and zero length crosslinkers [30,31]. Non-zero length crosslinkers, such as glutaraldehyde and polyepoxides, are bi- or multi-functional molecules that work by bridging the groups between adjacent polypeptides like free carboxylic acid/amino/hydroxyl groups. Whereas the zero-length crosslinkers like ethyl-3 (3dimethylaminopropyl) carbodiimide (EDC) work by forming covalent bonds between the reactive groups like carboxylic acid and amine groups that are available in proximity in the polypeptide chains. 

### 2.1. Natural Crosslinkers

The crosslinkers or chemical compounds obtained from natural sources like plants are categorized as natural crosslinkers. The commonly used natural crosslinkers are as follows:

#### 2.1.1. Genipin

Genipin is a natural plant product extracted from the fruit of *Gardenia jasmonoides* Ellis [32]. Genipin reacts with primary amines (Figure 1a,b), in particular the alpha or *N*-terminal amine and epsilon amine in the side chains of lysine in proteins like collagen, forming genipin-amino group monomers, which subsequently results in intermolecularly crosslinked collagen proteins. The efficiency of genipin crosslinking is almost similar to that of the aldehydes, but they are relatively non-toxic compared to glutaraldehyde which is toxic and leads to calcification [33,34]. Owing to the crosslinking efficiency and comparatively less toxicity, genipin has proved to be useful for producing corneal lens implants [35] and for the repair of intervertebral disc annulus damages [36]. Fessel et al. have investigated and proved the ability of genipin to arrest the propagation of tendon tears by an in vitro model and restoration of normalcy in tissue strains [37]. Later, Fessel et al. reported the effects of genipin crosslinking with respect to dosage and time on cell viability and cellular biomechanics for tendon repair applications [38]. Genipin of concentrations ~5 mM or slightly lower has proven to be successful for the rapid crosslinking of tissues ensuring cell viability in subpopulations of cells. For the clinical application of tendon repair, in situ cross-linking with genipin of a 5 mM dosage for a 72 h duration was concluded to be rational for in vivo studies wherein rapid restoration of tissues with biomechanical properties was achieved with a level of cytotoxicity and cell survival that was good enough to revive and stabilize the tissues at post crosslinking. Genipin crosslinking has also been utilized to form biopolymeric materials from collagen, gelatin, elastin and silk with biocompatibility, regulatable swelling and mechanical characteristics. These crosslinked materials find applications in drug release [39] and tissue repair/regeneration [40]. An interesting study has reported the exploitation of genipin crosslinking in developing a nano-vaccine based on functional fluorescent ovalbumin and chitosan nano-particles [41]. Genipin crosslinking, along with its fluorescence, ensures the tracking of antigen delivery in vivo. Also, using this, abundant antigens could be crosslinked without the need for additional carrier molecules. This nano-vaccine was found to induce antigen-specific immune responses, thereby serving as a novel model for vaccine delivery. Another recent report substantiated the applicability of genipin as a crosslinking agent for the formation of edible sodium caseinate films [42] with desirable mechanical properties, including tensile strength, Young’s modulus, and break elongation, along with the reduction of water vapor permeability in this edible protein film. The tunability of hydrogel structures and viscoelastic properties was demonstrated [43] by chronological crosslinking of silk fibroin proteins by physical as well as genipin crosslinking. Physical crosslinking of silk fibroin yields a β-sheet structure in the presence of high-pressure CO_2_. Crosslinking silk fibroin with genipin before β-sheet formation hinders the gelation process, in which the molecular weight increases affecting the morphology of the protein hydrogel and decreasing the gel stiffness. But crosslinking using genipin post-gelation was reported to increase the gel stiffness when the amorphous portions of the proteins are linked.

#### 2.1.2. Citric Acid

Citric acid is one of the commonly used polycarboxylic acids, which can crosslink with adequate safety, efficiency and cost-effectiveness. Due to the lack of knowledge on the mechanism of interaction of citric acid with the biopolymers or macromolecules, it had controversial existence as a crosslinker and plasticizer. However, crosslinking of proteins with citric acid was attempted by researchers by varying the pH conditions. Gliadin fibers were reported to be crosslinked using citric acid at various pH conditions and were found to result in tensile strength enhancement of up to 40% and tensile elongation of up to 100% [44]. The mechanism of citric acid crosslinking was reported for the first time with a model protein gliadin from wheat gluten, along with their kinetic parameters, performed at low temperatures in a neutral aqueous medium [45]. This study showed that more than one carboxyl group of a citric acid molecule reacted with the protein under aqueous conditions in the temperature range of 50–75 °C and pH 6.8. Nucleophilic substitution occurs between carboxyl groups in citric acid and the nucleophilic amine groups in one of the proteins (protein 1). Crosslinking also occurs with more than one carboxyl group of citric acid reacting with two/three proteins (proteins 1, 2 and 3) which could result in either intra-/inter-crosslinking (Figure 2). Nucleophilic substitution occurs by the attack of a partial negative charge on the nitrogen of NH_2_ and the partial positive charge on COO^−^, resulting in amide bond formation at increased pH conditions. This mild crosslinking with citric acid was anticipated to speed up the efficient enzyme immobilization process for the large-scale generation of protein-based products. The utilization of banana fibers and wheat gluten to develop films crosslinked with citric acid [46] showed the potential of agricultural residues to be converted into bio-based products that could serve as alternates for plastic-based packaging materials. Citric acid crosslinking aided in promoting the mechanical properties by enhancing film strength from 3.5 MPa to 13 MPa, as well as a reduction in water absorption from 500 to 200%. These films possessed higher strength compared to that of glutaraldehyde crosslinked films. The citric acid crosslinking was also applied in fabricating edible emulsions composed of whey protein isolates [47], in which citric acid and CaCl_2_ act as cold-gelling agents. More recently, fish gelatin and agar were crosslinked into films using citric acid, and their physical, chemical, mechanical and optical properties after thermal treatment at two different temperatures, 90 and 105 °C, were investigated [48]. The degree of crosslinking was found to be relatively more in those films treated at a temperature of 105 °C compared to those treated at 90 °C. Also, these thermally (both 90 and 105 °C) treated, citric acid crosslinked fish gelatin-agar films were reported to possess improved UV light barrier properties and are suitable for applications in food packaging and the development of other sustainable products. 

#### 2.1.3. Nordihydroguaiaretic Acid

Nordihydroguaiaretic acid is a natural polyphenol compound obtained from the plant source [49], creosote bush, which contains two reactive ortho-catechol functional moieties. Nordihydroguaiaretic acid has proven to be an anticancer and antioxidant compound [50], apart from being a natural chemical crosslinker [51]. Mechanical properties like the tensile strength and hardness of collagen crosslinked with nordihydroguaiaretic acid were found to be directly dependent on the concentration of the crosslinker. The nordihydroguaiaretic acid, upon autooxidation at neutral/alkaline pH conditions, is known to produce reactive quinones, which then couple together to form aryloxy free radicals resulting in bisquinone crosslinks at each end. This continues to generate a large network of crosslinked bisquinone polymer in which the collagen fibrils get embedded, forming a firm bio-stable matrix. This nordihydroguaiaretic acid treatment/crosslinking does not form crosslinks with amino acid side chains of collagen [52] (Figure 3). Nordihydroguaiaretic acid was reported to be used to crosslink gelatin to form hydrogels [51] with biocompatibility, thermal and mechanical stability, along with potential applications in the surgical field. Here, the crosslinking of gelatins prevented the solubilization of hydrogels in chaotropic agents and enhanced their thermal stability to above 80 °C. Also, it was concluded that the crosslinked 5% gelatin hydrogel was stable up to a temperature of 52 °C. Recently, etch and crosslink technique was followed with a bio-modified etchant nordihydroguaiaretic acid [53] at a low concentration range of 0.5 or 1% to achieve a stable, enzyme-resistant, mechanically strong and durable dentin collagen matrix. More recently, researchers in this field patented their novel methods of preparing nordihydroguaiaretic acid polymerized collagen fibers [54], along with the method of preparation of collagen fibers and medical devices, including ligament bio-prosthesis and other implantable devices with desirable tensile strengths.

#### 2.1.4. Procyanidins

Procyanidins are one of the important flavonoids currently researched for varied applications, including cardio protection [55], chemoprevention of colorectal cancer [56], inhibition of metalloproteinases activity (MMP-2 and MMP-9), rendering cancer chemoprevention [57], inhibition of in vitro human LDL oxidation [58], inhibition of advanced human prostate tumor growth [59], angiogenesis and so on. These are oligomers or polymers containing flavan-3-ol like epicatechin or catechin [60,61], and they are naturally found in plant metabolites, fruits, vegetables, seeds, nuts, flowers and barks. Procyanidins were found to crosslink collagen by forming new hydrogen bonds, as shown in Figure 4. Procyanidins are also reported to crosslink decellularized porcine heart valves without mineral deposition or calcification in simulated body fluids [62]. It was concluded that procyanidins inhibited osteo-differentiation and suppressed alkaline phosphatase activity, which ultimately blocked calcification of the crosslinked matrix, suitable for implantation. Similarly, procyanidins were also reported to crosslink with the protein elastin along with inhibition of calcification [63]. This anti-calcification could be due to several factors like inhibition of inflammatory factors/cell secretion, resistance to elastase activity, or inhibition of initiation of mineral deposition by elastin. These procyanidin-crosslinked elastins or aortic tissues were reported to possess high pore volume, large pore size in the range of 10–300 μm and high porosity of 75.1% suitable for the in vivo recellularization of scaffolds. Also, these crosslinked scaffolds exhibited anti-inflammatory, anti-thrombus activity along with hemocompatibility, suitable for scaffolds in tissue engineering applications. A new method of crosslinking involving procyanidins and glutaraldehyde was developed to improve the mechanical properties and bio-stability and reduce calcification and cytotoxicity in vascular extracellular matrices [64]. Procyanidins improvised the glutaraldehyde crosslinking by means of eliminating calcification and cytocompatibility, along with the enhancement in tensile strength and other mechanical properties. This method of co-crosslinking with procyanidins and glutaraldehyde was utilized in blood vessel tissue engineering.

#### 2.1.5. Tannic Acid

Tannic acid is a specific type of tannin or plant polyphenol, made up of D-glucose gallic acid ester with multiple phenolic hydroxyl groups and aromatic rings. It is available widely in fruits, grains, seeds of legumes and in beverages like tea, wine, cocoa, etc. [65]. Tannic acid possesses a relatively high molecular weight and can crosslink with proteins, carbohydrates, and other biomacromolecules [65]. It has the ability to interact with the protein collagen and is employed in conventional leather tanning methods [66]. The reaction between tannic acid and the amino groups of proteins/peptides is depicted in Figure 5. The mechanism of crosslinking with collagen involves the formation of hydrogen bonds and hydrophobic interactions [67]. A recent study demonstrated the exploitation of tannic acid in systemically delivering proteins/peptide therapeutics [9] to the heart, an existing challenge due to its complex physiology. Tannic acid-modified protein drugs would selectively bind to extracellular matrices, collagen or elastin but not to endothelial glycocalyx. These tannylated proteins could penetrate through the endothelial layer to the myocardial extracellular matrix to aid in specific drug delivery to the heart and hence to treat heart-related diseases. A tannylated fibroblast growth factor delivered to an injured myocardium of a rat model was found to enhance heart function. Tannic acid-protein cross-linkages were also efficiently exploited in developing capsules of tannic acid-bovine serum albumin [68] with controlled release of drugs supported by enzyme-specific degradation, where it remains intact with trypsin (hydrolysis reaction) but breaks with α-chymotrypsin (cleavage of carbonyl side of peptide bonds). This encapsulation technique has potential applications in the site-specific controlled release of drugs. Similarly, the multilayer assembly or micro-encapsulation of protein and tannic acid were developed [68] for its application in the oral delivery of bioactive functional foods. The natural chemical crosslinkers mentioned above, along with specific applications, are summarized in Table 1.

### 2.2. Synthetic Crosslinkers

Several synthetically generated chemicals are also used for crosslinking proteins. They are discussed as follows.

#### 2.2.1. Carbodiimide Agents 

The carbodiimide compound, 1-ethyl-3-(3-dimethylaminopropyl)-carbodiimide (EDC), is a water-soluble and most frequently used chemical reagent for crosslinking proteins by the formation of an amide bond as shown in Figure 6 along with step by step reactions involved for crosslinking of collagen using EDC. However, the crosslinking efficiency was influenced by various parameters, including pH, the nature of the solvent, stoichiometry, and side reactions. This method of crosslinking was reported to influence the structure and activity of the resulting crosslinked protein [74]. The hydrolyzed forms of collagen and gelatin proteins were reported to be modified and crosslinked with EDC in the formation of drug conjugates for anticancer treatment [75]. The influence of pH in EDC-gelatin protein crosslinking was investigated, and it was inferred that varied results were observed depending on the reaction pH and gelatin concentration [76]. No crosslinking occurred between EDC and gelatin when the protein concentration was 4 mg/mL, and the pH was 3 and 7. While at pH 5 complete reaction was found to occur. Also, the conditions under which EDC-induced protein degradation occurred were demonstrated [76] with gelatin-EDC crosslinking. Almost complete degradation was reported at pH 7, with excess EDC in the absence of amino-reactive groups at high temperatures. Moderate degradation was observed at pH 8.5, with excess EDC in the presence of reactive amino groups, whereas at pH 5–7, with excess EDC along with reactive amino groups, little or almost no protein degradation was reported. Collagen scaffolds used for various tissue regeneration purposes were usually crosslinked with EDC in the presence of N-hydroxysuccinimide (NHS). Although these crosslinked collagen scaffolds offer adequate biocompatibility and low cytotoxicity, the crosslinking of EDC/NHS with collagen was found to affect cell adhesion [77]. Integrin-mediated cell adhesion was found to occur by the binding of integrin to the cell adhesive motifs (GxOGER) on collagen [77] by means of a reaction between carboxylate anion on the side chain of glutamic acid of collagen in the presence of Mg^2+^ions of integrin. A similar reaction mechanism was found to occur during EDC/NHS crosslinking, i.e., carboxylate anion groups are required to crosslink the adjacent amine groups of lysine side chains of collagen motifs to form a peptide bond. Thereby, excessive crosslinking with EDC/NHS led to the unavailability of carboxylate anion on the collagen motif for integrin-dependent cell adhesion. This study [71] has proved to have a significant role in tuning the extent of crosslinking by the collagen scaffolds with EDC/NHS to 10% of the conventional conditions [78], which will restore the native cell adhesion properties. The electrospun gelatin nano-fibrous membranes chemically modified by carbodiimide with ethanol/water as co-solvents [79] were reported to regulate the structure-property relationships and resulted in biocompatible (both in vitro and in vivo) materials with thermal and biological stability suitable for potential ophthalmic applications.

#### 2.2.2. Epoxy Compounds

Epoxy compounds, in general, contain several epoxy functional groups that can react with amino, carboxyl and hydroxyl groups and be extensively utilized to crosslink proteins. Epoxy compounds are being used for the preservation of biological tissues like porcine arteries, and proteins, like collagen, via crosslinking. To understand the crosslinking reactions between proteins like collagen with the epoxy compounds, a water-soluble form of compound 1,4-butanediol diglycidyl ether (BDDGE) was chosen, and its influence on the extent and rate of crosslinking was explored [80]. The rate of crosslinking was relatively low compared to that of the glutaraldehyde process. The crosslinking reaction performed under basic conditions was reported to have proceeded by linking the amine groups of hydroxylysine residues of collagen protein that further resulted in a stiff/rigid material, while under acidic conditions, crosslinking occurred through carboxylic acid (COOH) groups of aspartic or glutamic acid residues of the collagen protein that formed a soft/flexible collagen material. Under basic conditions, nucleophilic attack occurs at the less substituted alkane of epoxide BDDGE with the amine group of a lysine residue in collagen, whereas under acidic conditions, nucleophilic attack occurs at the more substituted alkane of epoxide BDDGE with the carboxyl group of aspartate/glutamate residues in collagen. Also, the masking of amine or carboxylic acid groups in collagen with a monofunctional reagent, glycidyl isopropyl ether, led to the formation of a very stiff crosslinked collagen material. Apart from these studies in the past, more recent research also involved the following epoxy compound, 1,4-butanediol diglycidyl ether (BDDGE), that possesses crosslinking ability without introducing any toxic effect. Electrospun gelatin fibers were reported [81] to be crosslinked with BDDGE, which enhanced the mechanical properties and preserved the morphology of the crosslinked gelatin fibers without causing toxicity. BDDGE has also proved to be an alternative to aldehyde based crosslinking, in forming bovine gelatin films along with D-sorbitol as a plasticizing agent, for use as biodegradable food packaging materials [82]. It was shown that BDDGE of 1% (*w*/*w*) with dry gelatin resulted in improved mechanical properties of the gelatin film along with a desirable level of water solubility and water vapor permeability that are required for making degradable food packaging materials. The ability of in situ crosslinking of gelatin nano-fibers with 1,4-butanediol diglycidyl ether (BDDGE) was also demonstrated [83], where gelatin fiber size can be regulated with the concentration of BDDGE. These crosslinked gelatin nano-fibers that support cell adhesion, proliferation and generation of extracellular matrices are used in skin regeneration. The detailed mechanism of crosslinking collagen protein by BDDGE is depicted in Figure 7.

#### 2.2.3. Polysaccharide Derivatives Containing Aldehyde Groups

Polysaccharide derivatives with aldehyde groups have proved to be efficient crosslinking agents for proteins [84]. The formation of a borate-diol complex in alginate dialdehyde [85], a polysaccharide dialdehyde crosslinker, links gelatin by Schiff’s base reaction [86]. This results in the formation of an in situ injectable hydrogel possessing bio-degradability and adequate anti-inflammatory or anti-oxidative stress responses. These crosslinked gelatin hydrogels provide cell attachment and migration, facilitating the treatment of osteoarthritis. Another finding reported that the degree of crosslinking between gelatin and oxidized cellulose nano-whiskers increased with the increase in the number of aldehydes [87]. It yielded a crosslinked gelatin hydrogel with increased rigidity, thermal resistance, a 150% increase in storage modulus and a relative reduction in water permeability. Also, the higher adsorption capacity of a protein, bovine serum albumin [88], on to dialdehyde cellulose crosslinked chitosan was demonstrated with an increase in the number of amino groups by the reaction of dialdehyde cellulose and chitosan. Dialdehyde micro-fibrillated cellulose, formed by surface modification of corresponding cellulose with periodate, was mixed with gelatin to form a composite hydrogel by Schiff base reaction in which the aldehyde group of dialdehyde cellulose was covalently bound to amine groups of the protein matrix of gelatin [89]. It was concluded that the pore size of the composite hydrogel could be modulated by changing the oxidation levels of dialdehyde micro-fibrillated cellulose. The mechanical strength of this composite hydrogel was reported to be extremely high, and the compression strength was enhanced 41-fold relative to that of pure gelatin or hydrolyzed collagen. The feasibility of another crosslinker, dialdehyde carboxymethyl cellulose formed by the oxidation of carboxy methyl cellulose, was demonstrated with proteins like gelatin [90] and other biopolymers like chitosan [91], which was cytocompatible, bio-degradable, thermally stable and well suited for biomedical applications like wound dressings. Pullulan is a natural polysaccharide found in the fungus *Aureobasidium pullulans* and was proven to be non-toxic, non-immunogenic, and bio-degradable bio-material that can find applications in specific drug/gene imaging and tissue engineering [92]. It mainly contains continuous maltotriose units linked by α-1,6 glycosidic bonds, and these maltotriose units are joined by two β-1,4 glycosidic bonds. The existence of different glycosidic bonds makes them exhibit unique physicochemical properties [93]. There are also reports stating that pullulan has been crosslinked and oxidized with sodium periodate (Figure 8) and blended with human-like collagen resulting in soft tissue fillers for skin reconstruction [94]. Some of the limitations of the pullulan hydrogels included inflammation, inadequate mechanical strength, and degradation. Further, these properties were enhanced by forming a composite hydrogel of pullulan-human-like collagen that was evaluated to be successful both in vitro and in vivo.

#### 2.2.4. N-hydroxysulfosuccinimide and Aryl Sulfonyl Fluoride

A hetero bifunctional cross-linker, N-hydroxysulfosuccinimide and aryl sulfonyl fluoride (NHSF), with a “plant-and-cast” cross-linking strategy was reported to target multiple residues of varying reactivity against nucleophilic amino acid side chains [95]. Here the hetero bifunctional cross-linker containing a highly reactive group, succinimide ester, was ‘planted’ at the lysine residue on the surface of the protein, followed by ‘casting’ of a less reactive aryl sulphonyl fluoride that reacts by a sulfur-fluoride exchange with the nearby weak nucleophilic side chains of Ser, Thr, Tyr, His, or Lys residues of the protein (Figure 9). It is significant to note that these sulfonyl fluorides can crosslink by specifically targeting the amino acid residues Ser, Thr, His, and Tyr that are inactive, which is not possible by other previously reported crosslinkers. Also, they were able to crosslink whole cell lysates of *E. coli* resulting in various interlinked peptides that could be analyzed by mass spectrometry. This proximity-dependent crosslinking [95] proved to possess the potential to identify enzyme-substrate interactions in live cells, as well as for protein structure identification with mass spectrometry.

#### 2.2.5. Quinone Methides

Quinone methides (QM) are known Michael acceptors for nucleophiles with versatile roles in chemical biology [96,97,98,99]. An unnatural amino acid, (2*R*)-2-amino-3-fluoro-3-(4-((2-nitrobenzyl)oxy) phenyl) propanoic acid (FnbY) capable of generating *para*-Quinone methides upon photoactivation, was reported to be genetically encoded in the proteins of *E.coli* and mammalian cells, and exhibited crosslinking ability towards multiple proximal nucleophilic amino acid residues, including Cys, Lys, His, Tyr, Trp, Met, Arg, Asn, and Gln, that were inaccessible to previously reported unnatural amino acids [100]. The photo-controllability of photo-crosslinkers and specific chemical reactivity of chemical crosslinkers were integrated together for the development of a series of photo-caged quinine methide [99] based small molecule cross-linkers and was reported to have crosslinked proteins by targeting multiple amino acid residues through Michael addition. It was proved to crosslink proteins in vitro in *E. coli* and mammalian cells. Initially, a heterobifunctional cross-linker denoted as ‘NHQM’ was fabricated by the reaction of an N-hydroxysuccinimidyl (NHS) ester and a photo-caged *ortho*-quinone methide, as shown in Figure 10. Upon adding proteins to this hetero functional crosslinker, NHS-ester reacts with the Lys residues exposed and plant o-quinone methide to adjacent residues. Then, on exposure to UV, the elimination of fluoride ions occurs, followed by a reaction of the o-quinone methide with adjacent nucleophilic side chains of amino acids in proteins. Also, the crosslinking was regulated by UV exposure, i.e., the crosslinking ability increased with an increase in the time of UV treatment. ‘NHQM’ was shown to crosslink the Lys of one peptide with other amino acid residues, like Gln, Lys, Glu, Ser, Arg, Asn, and Asp of another peptide. The crosslinker’s flexibility was found to increase by using another quinine methide compound, ‘NHQM3C’ [99], that was formed by the addition of three methylenes at the spacer of ‘NHQM’ [99]. Here the Lys residue of one protein was shown to be crosslinked with the nucleophilic groups of amino acid residues, Gln, Glu, Thr, and Tyr of another protein upon photo activation at 365 nm. This has expanded the use of natural amino acids, targetable with only one crosslinker. For protein crosslinking to occur inside the cells, a homo bifunctional crosslinker, ‘HoQM,’ was designed with photocaged *o*-QM at both ends, replacing the NHS-ester so that the chemical reactivity would be eliminated before photoactivation and could enter the cells without causing any toxic effects. ‘HoQM’ treatment with *E.coli* cells crosslinked the intracellular expressed protein to form dimers upon UV exposure. The same was not feasible with previously reported ‘NHQM.’ ‘HoQM’ [99] was reported to crosslink proteins in the live cells of both *E.coli* and mammalian cells.

#### 2.2.6. β-[tris(hydroxymethyl) phosphino] Propanoic Acid (THPP)

β-[tris(hydroxymethyl) phosphino] propanoic acid (THPP) was initially reported as a crosslinker for the generation of elastin-like protein materials [101,102]. This acts as a trifunctional crosslinker that covalently crosslinks the primary and secondary amine groups forming a formaldehyde intermediate, succeeded by Mannich-type condensation of amine-formaldehyde and immonium-phosphorus ions, at physiological conditions in an aqueous medium [103,104,105]. These crosslinked proteins were found to rapidly result in hydrogels [106] non-cytotoxic and possessed the ability to encapsulate cells [101,102,107]. Since this crosslinker involved complex reactions for synthesis, another crosslinker, tetrakis (hydroxymethyl) phosphonium chloride (THPC), with high structural similarity, was developed [103]. THPC crosslinks by reacting with the amine groups of amino acids, like lysine residues with two primary amine groups, for generating crosslinked protein hydrogel-like materials. A schematic representing the mechanism of crosslinking is shown in Figure 11. This reaction of THPC with amino acids like lysine led to relatively more formaldehyde intermediate formation in comparison to other amino acids like glycine, cysteine or proline. This was due to the two primary amine groups in lysine residue, which resulted in favorable reactions with an intermediate formaldehyde that triggered the replacement of hydroxyl groups in THPC with lysine residues of the protein to be crosslinked.

#### 2.2.7. Protein Partners-SpyTag/SpyCatcher System

The SpyTag/SpyCatcher system was developed by a covalent reaction between the protein partners acquired from the fibronectin-binding protein (FbaB) of Streptococcus pyogenes [108]. Here the covalent linkage occurs by a specific isopeptide bond between the amino acid residues Asp-117 of SpyTag and Lys-31 of SpyCatcher [109]. This peptide or amide bond (Figure 12) was found to form autocatalytically by combining a peptide (SpyTag, 1.1 kDa) and a small protein (SpyCatcher, 12 kDa) [108]. This protein-peptide crosslinker was shown to generate elastin-like protein polypeptides with various orientations of the protein secondary structures [110]. The SpyTag–SpyCatcher system finds potential applications in the formation of self-assembled protein nano-materials with biological functionalization [109,110,111,112,113]. This provides a new platform for crosslinking proteins, resulting in protein-based materials with potential use as scaffolds for tissue engineering applications [109].

The chemical crosslinking methods using synthetic crosslinkers mentioned above, along with significant applications, are summarized in Table 2.

#### 2.2.8. Miscellaneous Crosslinkers

There are several other crosslinkers that find potential applications as bioconjugates, target/ functional group-specific linkers, nano-particle preparation, etc. Recently several novel maleimides that are rationally designed and are known to overcome the shortcomings of classical maleimides have been reported in the literature. The very good reactivity and synthetic accessibility offered by these maleimides towards bioconjugation make them powerful tools for protein crosslinking, particularly for drug conjugation to antibodies. The mechanism of crosslinking proteins using maleimide esters is depicted in Figure 13.

Similarly, numerous photo affinity probes have also been recently reported and are known to be involved in protein-protein interactions. A bifunctional amino acid containing a terminal alkyne group and a diazirine as a photo-crosslinker was utilized for bio-orthogonal tagging (A bifunctional amino acid to study protein–protein interaction [114]. The most challenging part of biomolecular crosslinking is to retain their activity even after crosslinking. This was made possible by following Click chemistry which is known to be highly selective and efficient in producing stable bio-orthogonal crosslinked products. This chemistry has been successfully demonstrated for the immobilization of biomolecules, nanofiber surfaces, and saccharides. Glutaraldehyde is another crosslinker that is known to immobilize amine-rich biomolecular substrates, as shown in Figure 14 [115].

## 3. Conclusions

Herein we have discussed various chemical methods of protein crosslinking. The applications of these strategies or methods to generate crosslinked protein networks or scaffolds, along with the limitations of each, are reviewed in detail, along with recent literature updates. The implementations of the generated biomaterials for applications in biomedical and food packaging industries are discussed, along with their advantages and limitations. A comparison of the toxicity generated and strength exhibited by materials created using natural agents over synthetic materials is elaborated. Moreover, chemical crosslinking methods employing natural agents are used to obtain covalent crosslinking as that obtained using synthetic agents without causing toxicity to the crosslinked material/film. Several applications, including scaffold materials, biomaterials for tissue engineering, drug delivery systems/vehicles, food packaging materials, edible protein films etc., have been achieved by chemical methods of protein-based crosslinking (Table 1 and Table 2).

## 4. Future Scope

Protein biomolecules possessing versatile features can be exploited for multiple applications. The extent of reinforcement is very much essential and needs to be achieved with better efficiency through novel crosslinking methods that can overcome the limitations of the presently available methods that result in protein-based materials/films which are non-toxic and biodegradable. With the constant demand in the bio-medical field for various protein-based scaffolds, tissue sealants, drug carriers and other bio-prosthetics, it is necessary for researchers to focus on creating novel protein-based materials that ultimately require protein crosslinkers. Biodegradable protein materials [116] could serve as replacements for the existing non-biodegradable synthetic polymers in packaging industries or other coating-based industries, and hence the environmental pollution due to synthetic polymer usage can be prevented.

Most of the existing protein crosslinking strategies, especially chemical crosslinkers, have the possibility of causing damage to protein structure/properties or the cells/tissues in end-use. Chemical crosslinking using synthetic crosslinking agents like EDC-NHS faces the major issue of cytotoxicity, and the removal of residual crosslinking agents is crucial. Natural chemical crosslinking agents, though they have the advantage of not conferring cytotoxicity, are known to possess a lower degree of crosslinking relative to that of synthetic crosslinkers. However, there are many other protein-based crosslinkers like leucine zipper domains or SpyTag/SpyCatcher systems and natural chemical crosslinkers that reinforce the protein polymers into materials without any toxic effects have always been the interest of many researchers. Specific protein-protein interactions with naturally available chemical compounds from plants or synthetic chemicals that can act like catalysts or zero-length crosslinkers ought to be explored further for better protein crosslinking efficiencies. Massive chances are still available to develop many more novel protein-based or natural/synthetic chemical crosslinkers to be utilized for various available proteins for the betterment of humankind.

## Data Availability

Not applicable.

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
