# Peer review of "Insights on Chemical Crosslinking Strategies for Proteins"

_molecules, 2022, doi:10.3390/molecules27238124_

Round 1

Reviewer 1 Report

The authors presented the paper "Insights on Chemical Crosslinking Strategies for Proteins"

1) Much more fresh 2-3 years paper should be presented in the Introduction section to show the area perspectives.

2) In the Introduction section, crosslinking strategies are discussed (lines 48, 69). It will well present a Picture, which can summarize the strategies.

3) Table 1 and 2. It will be good to present some limitations as a new column. Moreover, Molecules is a multidisciplinary journal. In this way, it will be excellent presenting for each crosslinking reagent structure and action mechanism. You have the picture only for some of them.

4) Some chemical crosslinkers are missed. See the Sigma catalog https://www.sigmaaldrich.com/DE/en/products/chemistry-and-biochemicals/chemical-biology/crosslinkers

maleimide-ester, diester, photoactivated (N3) for some specific linking, click reactions

glutaraldehyde for protein nanoparticle production

Minor comments

1) Graphical Abstract in Molecules J. is not usually presented in the paper. It is uploaded as a separate file.

2) Some picture (as graphical abstract) of the best crosslinkers may be presented in the Sections 3 or 4.

Author Response

Reviewer 1:

The authors presented the paper "Insights on Chemical Crosslinking Strategies for Proteins"

1. Much more fresh 2-3 years paper should be presented in the Introduction section to show the area perspectives.

Response: As suggested recent references are now cited in the revised manuscript.

2. In the Introduction section, crosslinking strategies are discussed (lines 48, 69). It will well present a Picture, which can summarize the strategies.

Response: We have included Scheme 1 depicting the various available functional groups in proteins / peptides along with examples that can be modified either physically, chemically or enzymatically.

3. Table 1 and 2. It will be good to present some limitations as a new column. Moreover, Molecules is a multidisciplinary journal. In this way, it will be excellent presenting for each crosslinking reagent structure and action mechanism. You have the picture only for some of them.

Response: The separate column listing the significant limitations of each crosslinker is now included in the revised manuscript. Also the mechanism of crosslinking for each molecule discussed in this manuscript is represented pictorially as figures.

4. Some chemical crosslinkers are missed. See the Sigma catalog https://www.sigmaaldrich.com/DE/en/products/chemistry-and-biochemicals/chemical-biology/crosslinkers - maleimide-ester, diester, photoactivated (N3) for some specific linking, click reactions - glutaraldehyde for protein nanoparticle production

Response: Yes we do agree that there are numerous crosslinkers available commercially as listed by the reviewer. But in this manuscript, we have discussed the most significant crosslinkers that are utilized for the formation of protein hydrogels. As recommended by the reviewer we have now included target specific crosslinkers for specialized applications and a brief account of the listed crosslinkers has now been included in the revised manuscript.

Minor comments

1. Graphical Abstract in Molecules J. is not usually presented in the paper. It is uploaded as a separate file.

Response: The graphical abstract is now removed from the main manuscript.

2. Some picture (as graphical abstract) of the best crosslinkers may be presented in the Sections 3 or 4.

Response: The mechanism of crosslinking is now included for all the molecules discussed in the manuscript as Schemes, Tables and Figures.

Reviewer 2 Report

The manuscript “Insights on Chemical Crosslinking Strategies for Proteins” by Brindha et al. presents a comprehensive review of different chemical compounds used for crosslinking of proteins. The review divides the chemical crosslinkers into natural and synthetic crosslinkers, highlighting the main crosslinkers of each category. The different crosslinking mechanisms, as well as the advantages and disadvantages of each family of crosslinkers, are well described.  The conclusions and future scope are also well stated. The only inconsistency in this well-written review is that the mechanism of action and structure of some crosslinkers is graphically shown, but not others. The structures of procyanidins and tannic acid are shown in the graphical abstract, but not in sections 2.1.4 or 2.1.5, respectively. Figures summarizing the structure and mechanism of action of these two natural crosslinkers will help in the understanding of these sections. Similarly, Figures containing the structures and mechanisms of protein crosslining of epoxy compounds and THPP synthetic crosslinkers should be included in sections 2.2.2 and 2.2.6, respectively.

Despite these minor issues, the review is worth for publication in molecules.

Author Response

1. The conclusions and future scope are also well stated. The only inconsistency in this well-written review is that the mechanism of action and structure of some crosslinkers is graphically shown, but not others.

Response: We thank the reviewer for the valuable suggestion. As suggested we have now included the mechanism for all the crosslinkers discussed in this manuscript.

2. The structures of procyanidins and tannic acid are shown in the graphical abstract, but not in sections 2.1.4 or 2.1.5, respectively. Figures summarizing the structure and mechanism of action of these two natural crosslinkers will help in the understanding of these sections.

Response: The structures and crosslinking mechanism of procyanidins and tannins are now included as figure 4 and figure 5 respectively in the revised manuscript.

3. Similarly, Figures containing the structures and mechanisms of protein crosslinking of epoxy compounds and THPP synthetic crosslinkers should be included in sections 2.2.2 and 2.2.6, respectively.

Response: The protein crosslinking mechanism using epoxy and tetrakis (hydroxymethyl) phosphonium chloride are shown in figures 7 and figure 11 respectively.